# Study on the Interface Microstructure of TaC/GCr15 Steel Surface Reinforced Layer Formed by In-Situ Reaction

**DOI:** 10.3390/ma16103790

**Published:** 2023-05-17

**Authors:** Jilin Li, Ruixue Li, Yunhua Xu, Zhuolin Liu, Le Chen, Yao Zhu

**Affiliations:** 1School of Material Science and Engineering, Xi’an University of Technology, Xi’an 710048, China; xuyunhua2023@163.com; 2Institute of New Materials, Guangdong Academy of Sciences, Guangzhou 510650, China; chenle6057@163.com (L.C.); zhuyaonihao@163.com (Y.Z.); 3School of Materials Science and Chemical Engineering, Harbin University of Science and Technology, Harbin 150080, China; 15353087157@163.com; 4School of Materials Science and Engineering, Central South University, Changsha 410083, China; liuzhuo0lin@163.com

**Keywords:** in situ reaction, TaC-enhanced layer, FIB microzone TEM, EBSD phase structure, microstructure and structure

## Abstract

In this study, a micro–nano TaC ceramic steel matrix reinforced layer was prepared by an in situ reaction between a pure tantalum plate and GCr15 steel. The microstructure and phase structure of the in situ reaction reinforced layer of the sample at 1100 °C and reaction time 1 h were characterized with FIB micro-section, TEM transmission, SAED diffraction pattern, SEM and EBSD. The phase composition, phase distribution, grain size, grain orientation and grain boundary deflection, phase structure and lattice constant of the sample were characterized in detail. The results show that the phase composition of the Ta sample is Ta, TaC, Ta_2_C and α-Fe. TaC is formed after Ta and carbon atoms meet, and the orientation changes in the X and Z directions. The grain size of TaC is widely in the range of 0~0.4 μm, and the angular deflection of TaC grain is not obvious. The high-resolution transmission structure, diffraction pattern and interplanar spacing of the phase were characterized, and the crystal planes of different crystal belt axes were determined. The study provides technical and theoretical support for further research on the preparation technology and microstructure of the TaC ceramic steel matrix reinforcement layer.

## 1. Introduction

With the development of modern science and technology, the working conditions of materials are becoming harsher. Meeting the performance requirements proposed by the actual production with traditional single materials has been difficult. Metal has high strength, high toughness and other properties, and ceramic materials have the advantages of high melting point, high hardness and good chemical stability [1]; taking into account the advantages of both ceramic-reinforced and metal-based composites, reinforced materials have been widely used in the automotive, aerospace, aviation, electronics, telecommunications and other fields and have a huge development prospect [2,3]. Most of the damage behaviors such as wear, corrosion, fatigue and fracture occur on the surface of the material [4]. According to statistics, the national economic loss caused by the surface failure of materials is huge [5,6]. By using materials with high hardness, wear and corrosion resistance to locally strengthen the surface of metal substrates, it is possible to reduce the costly effects of overall strengthening and to take into account the toughness of the material core. Steel is the most used engineering alloy in the industry, due to its high strength and toughness, easy processing and low cost; it is widely used in various fields. However, its service life is greatly reduced due to its low hardness, wear resistance and corrosion resistance, which limits its wider application [7,8]. Ceramic materials such as carbides, oxides, borides and nitrides with high hardness, high stiffness and refractory characteristics are selected as reinforcements [9,10,11,12]. The in situ reaction preparation of reinforcement layer technology has the advantages of no pollution on the material surface; good bonding with the substrate; controllable particle size, composition, shape and distribution; and lower cost, etc. [13,14,15,16,17]. Jia et al. [18] prepared titanium–silicon–carbon ceramic materials by the hot-press firing method with titanium powder, silicon carbide powder, tantalum carbide powder and graphite as raw materials, and found that tantalum carbide had good chemical compatibility with titanium–silicon–carbon. The introduction of tantalum carbide not only improved the densification of titanium–silicon–carbon ceramic materials, but also improved the mechanical properties of ceramic materials; however, excessive amounts would produce bad effects.

In this study, high-carbon chromium GCr15 bearing steel was used as the substrate and TaC as the reinforcement, and the reinforcement layer was formed by in situ solid diffusion. The surface of GCr15 bearing steel is prone to cracking, wear and corrosion after a long service period in a high-speed environment [19], and surface strengthening can be performed to improve its service life. TaC, as a carbide of the fifth paragenetic transition metal [20,21], has the advantages of high hardness [22], high modulus [23] and good wear resistance, stable high-temperature properties and strong resistance to acid and alkali corrosion. It is commonly used as the material of choice for the preparation of reinforcement layers.

## 2. Materials and Methods

### 2.1. Raw Materials Preparation

GCr15 bearing steel is commonly used as a high-chromium bearing steel, with high and uniform hardness and hardenability obtained after heat treatment. It has excellent wear resistance, high contact fatigue strength, good dimensional stability and corrosion resistance, but it has medium cold-deformation plasticity and is widely used in the production of various bearing rings and rolling elements. In this study, the GCr15 bearing steel, which provides the required C atoms through a diffusion reaction, is selected as the substrate, and a TaC-reinforced layer is formed by in situ reaction with a pure Ta plate.

Tantalum is a strong carbide-forming element that reacts with the C atoms provided by the substrate to produce carbide-reinforced particles. In this study, high-purity tantalum plates (≥99.9%) were used to provide Ta atoms needed for the in situ reaction process. The specific grades and compositions of the materials used are shown in Table 1.

### 2.2. Preparation Methods

The GCr15 bearing steel was selected as the experimental material and processed into a cylindrical material of ø20 × 22 mm by wire-cutting. A pure tantalum plate with a thickness of 0.5 mm was processed into a round piece of ø20 mm. A schematic diagram of the GCr15 steel and tantalum plate specimens is shown in Figure 1.

The GCr15 bearing steel and tantalum plate specimens were polished with sandpaper of different grits, from 240 mesh to 2000 mesh, then put into anhydrous ethanol and cleaned with ultrasonication until the surface was clean, then finally blown dry and set aside.

The preparation process of the TaC-reinforced steel-based surface local reinforcement layer is shown in Figure 2, where the two pre-treated raw materials are closely laminated to form a preform, the preform is wrapped with graphite paper and placed in an isostatic graphite mold with an inner diameter of ø20 mm and an outer diameter of ø60 mm, and the in situ reaction is carried out in a solid-state diffusion reaction pressure sintering furnace, so that the Ta and C elements undergo a solid-phase diffusion reaction, and the sample preparation of a micro–nano TaC ceramic particle-reinforced layer is realized.

Solid diffusion in situ reactions were performed at temperatures of 1050, 1100 and 1150 °C, held for 1, 2 and 3 h, respectively. The performance of samples at 1100 °C 1 h was the best. Thus, the process parameters of the TaC/α-Fe surface micro–nano ceramic-reinforced layer prepared by in situ reaction were the following: a temperature of 1100 °C for 1 h and a constant pressure of 30 MPa in the system of a Ta plate and GCr15 bearing steel. The process curve of the reaction preparation process is shown in Figure 3.

The specific operation was to put the prepared perform into the diffusion reaction sintering furnace, vacuum the furnace to 10^−3^ Pa, and then pass argon gas for atmosphere protection (the argon gas pressure was in the range of 5–12 KPa and the temperature was kept at 1100 °C for 1 h). During the reaction process, the constant pressure was 30 MPa. This was followed by cooling to 800 °C for 30 min (the heat preservation was sufficient to release the stress in the reinforced layer), and finally, cooling the furnace to room temperature.

### 2.3. Experimental Instruments and Equipment

In this study, the samples were prepared with a diffusion reaction pressure sintering furnace (model: ZT-40-21Y). A JSM 7200F was equipped with an EDAX Velocity Super for EBSD (the sample coordinates were in the transverse direction, radial direction, and normal direction, with a sample inclination angle of 70°) to determine the grain orientation characterization and analyze the orientation relationship between TaC growth and the matrix. The interface structure of the TaC-reinforced layer and matrix was characterized by a focused ion beam (FIB, FEI Helios Nano Lab 600i, USA) and a transmission electron microscope (TEM, FEI Talos F200X, USA).

## 3. Microstructure and Microstructure Analysis

### 3.1. EBSD Crystal Plane Orientation Analysis

The phase composition of the Ta/TaC and TaC/Fe interface was characterized and analyzed with EBSD. EBSD technology is based on the difference in the crystal structure, space group and corresponding different parameters to identify different crystal structure phases, to obtain important material characterization information such as phase composition, phase distribution, grain size, grain orientation and grain boundary. Table 2 lists the crystal structures and corresponding parameters of different types of carbides.

Preparation of TaC sample EBSD sample was as follows: The TaC sample reacted at 1100 °C for 1 h and was polished step by step with 200–3000 mesh sandpaper. Vibration-polishing was performed at a frequency of 50 Hz on the VibroMet 2 vibration-polishing machine. An electron microscope equipped with an EDAX Velocity Super in JSM 7200F was used for observation. The EBSD shooting area of the TaC sample is shown in Figure 4. The figure shows the thickness of the reaction-enhanced layer and its interface bonding morphology.

Figure 5 shows the EBSD morphology and phase distribution of the sample. The EBSD interface and phase distribution of the sample interface are analyzed in the figure. Figure 5a is the BSE diagram of the selected interface study area. To determine the phase distribution, the EBSD phase diagram was obtained. In Figure 5b, the phase distribution corresponding to the BSE morphology can be observed. The red area is tantalum metal, the green area is a small amount of Ta_2_C dispersedly distributed, the blue area is the main TaC, and the orange area is α-Fe. It can be seen from the diagram that the strengthening layer is mainly composed of TaC; Ta_2_C is dispersed in the strengthening layer. The serrated bonding interfaces (left) are the bonding layer and the Ta plate, the layered bonding interfaces (right) are the strengthening layer and α-Fe. 

Figure 6a–c are grain distribution maps in different directions. Figure 6d is the antipole diagram of different phases. From Figure 6a, it can be seen that in the X direction, the metal tantalum is mostly blue, so the main orientation is in the [111] direction; in the X direction, α-Fe is mostly blue, so the main orientation is also in the [111] direction. However, the TaC and Ta_2_C grains are fine and have no obvious orientation, and most of their grain orientation cannot be judged. It can be seen from Figure 6b that the metal tantalum in the Y direction is mostly purple, so the main orientation is in the [122] direction; in the Y direction, α-Fe is mostly orange, so it is mainly oriented in the [001] direction, but the TaC and Ta2C grains are small and have no obvious orientation, so it is impossible to judge the orientation of most grains. In the Z direction, shown in Figure 6c, the metal tantalum is mostly green, so the main orientation is in the [101] direction; in the Z direction, α-Fe is mostly purple, so the main orientation is in the [122] direction, but the TaC and Ta_2_C grains are small and have no obvious orientation, so it is impossible to judge the orientation of most grains.

Figure 7 is the EBSD pole figure (PF) and inverse pole figure (IPF) results of Region-I and Region-II. From Figure 7a, it can be seen that TaC has the highest intensity and more distribution on the {001} crystal plane family. From Figure 7c, it is given that the intensity of the crystal direction [101] in the X direction is the highest, the intensity of the crystal direction [122] in the Y direction is the highest, and the intensity of the crystal direction [112] in the Z direction is the highest. Combining the pole figure and the IPF diagram, it is judged that the TaC {001} crystal plane grows along the crystal direction [101] in the X direction. It grows along the crystal direction [122] in the Y direction and along the crystal direction [112] in the Z direction. From Figure 7b, it can be seen that Ta has the highest strength on the {001} crystal plane family. From Figure 7d, it can be seen that the strength of the crystal direction [111] in the X direction is the highest, the strength of the crystal direction [122] in the Y direction is the highest, and the strength of the crystal direction [313] in the Z direction is the highest. Combined with the pole diagram and IPF diagram, it is judged that the Ta {001} crystal plane grows along the crystal direction [111] in the X direction, along the crystal direction [122] in the Y direction, and along the crystal direction [313] in the Z direction. Through analysis, the crystal orientation and crystal orientation relationship of the above phases is summarized as follows:
TaC: X:[101] Y:[122] Z:[112] {001} (1)

Ta: X:[111] Y:[122] Z:[313] {001}(2)


The EBSD results of the Ta samples confirm the phase composition and crystal orientation relationship of the Ta/TaC-Fe interface. Because tantalum carbide is formed in the process of diffusion and in situ reaction, the nucleation of tantalum carbide depends on both external and internal factors. The external factors are in situ reaction temperature, reaction time and cooling conditions. The internal factors include the crystal structure of tantalum, the content of ion defects and the critical condition of carbide nucleation. The above factors will affect the nucleation of tantalum carbide, making tantalum carbide nucleate on the habit plane or the specific crystal plane of the matrix, which is an important reason for the orientation of the crystal plane. In addition, the growth or coarsening stage of tantalum carbide after nucleation is affected by the concentration of C atoms and Ta atoms, the surface energy of different crystal planes, interface energy and grain boundary migration ability. The growth of tantalum carbide can proceed preferentially along one or several specific directions, resulting in crystal orientation.

Figure 8 is the histogram of grain size distribution. The grain size is mainly less than 0.2 μm, the Ta grain size is 0~0.2 μm, the Ta_2_C grain size is between 0 and 0.2 μm, and the TaC grain size is widely distributed between 0 and 0.4 μm. All grains are generally small, and Ta_2_C is fine and dispersed in the strengthening layer, so the growth of Ta_2_C grains is limited, while most of the strengthening layer is TaC, so there are some large grains in the TaC grains.

Figure 9 shows the grain boundary distribution of Ta, TaC, Ta_2_C and α-Fe phases. The red line is 0~2°, the yellow line is 2~5°, the green line is 5~10°, and the blue line is 10~15°. Among them, the angle between 5 and 15° is the main angle, and it mostly exists between TaC/Ta_2_C and TaC/TaC grains. The diagram does not calibrate the high-angle grain boundary. Figure 10 shows the grain boundary deflection angle of different phases. Generally, a grain boundary greater than 15° is called a high-angle grain boundary (HAGB), and a grain boundary less than 15° is called a low-angle grain boundary (LAGB) [24]. Figure 10 is judged by different relatively high-angle grain boundaries. The content of TaC grains in high-angle grain boundaries and low-angle grain boundaries is equivalent. It is generally believed that the high-angle grain boundary deflection angle is caused by grain deformation or grain recrystallization caused by stress, while the high-angle grain boundary deflection angle of TaC grains and Ta_2_C grains in this study is caused by the lack of preferred orientation during grain growth.

### 3.2. Analysis of TEM Microstructure and SAED Diffraction Pattern

To further determine the size of the products and reinforcements in each region of the gradient composite layer reinforced with micro and nano TaC ceramic particles, focused ion beam (FIB) selective sectioning and HRTEM transmission analysis were performed on each region. The TaC sample reacted at 1100 °C for 1 h and was characterized by combining FIB and TEM, and its characterization was analyzed and processed.

The cross-sectional structure of the tantalum carbide diffusion enhancement layer on the surface of GCr15 prepared at 1100 °C for 1 h was characterized with TEM, as shown in Figure 11. The tantalum carbide particles in the diffusion enhancement layer have fine grains, unclear grain boundaries, and a large number of dislocation entanglements, which is consistent with the conclusion that the residual stress in the sample is largely based on the FIB sample-cutting analysis. Performing a spot scan of Figure 11a, it is seen that the composition difference between the spot scan results at Spot 1 and Spot 2 is very small. The two grains are the same type of particle. From the content of the C element, it can be seen that the position away from the Ta layer has a small content of C, which conforms to the diffusion rule. It is necessary to conduct a diffraction pattern analysis to determine whether the grain is TaC or Ta_2_C. It can be seen from Figure 11b that there is an obvious pearlite structure at the matrix, with relatively smooth interface bonding, but the grain boundary is still not obvious. The dot scan results at Spot 3 and Spot 4 show that the C, Ta, and Fe elements vary greatly on both sides of the interface, with large sizes of Ta and Fe atoms and weak diffusion ability. Ta is a strong carbide element that binds to C elements more strongly than the matrix, resulting in a higher content of C elements in the diffusion reaction layer than in the matrix (C source). The TaC diffusion reaction layer/substrate interface can only be observed to have holes caused by ion thinning, and there are no obvious defects at the bonding interface. This indicates that the reinforcement has an excellent microscopic interface, but residual stress at this location needs to be eliminated.

Figure 12 is a high-resolution TEM photograph of the Ta/TaC diffusion reaction layer interface. As shown in Figure 12, the equiaxed particles near the tantalum grain interface are hexagonal close-packed Ta_2_C, and the Ta_2_C crystal plane spacing on the (100) is 0.272 nm. From Figure 12d–f, it can be seen that the grain interface between Ta_2_C particles is not obvious, and there is a large number of dislocations between the particles. Observing Figure 12e,f of the FFT conversion, it can be found that the upper dislocation is obvious on the (100) and (010) crystal planes. Due to the large number of Ta atoms on the side near the tantalum interface and the limited number of C atoms diffusing thereto, the grains are mostly Ta_2_C particles.

Figure 13 is a TEM photograph of the TaC/matrix interface and the interface between TaC particles. From Figure 13b, it can be observed that the TaC/matrix interface is well bonded, the interface is clear, and there are no obvious dislocations around the interface. From Figure 13c, it can be observed that there is a significant lattice distortion phenomenon between TaC particles at the interface, and the crystal surface near the interface is distorted, which is caused by a large number of dislocations and element diffusion at the interface.

Figure 14 shows a high-resolution TEM image of the TaC/matrix diffusion reaction layer interface. From Figure 14, it can be seen that the TaC has a face-centered cubic structure (fcc), and the α- Fe has a body-centered cubic structure (bcc). From Figure 14a, it can be seen that there are obvious holes at the interface of the Ta/TaC diffusion reaction layer, resulting from excessive residual stress at the interface and bending deformation during ion thinning, leading to the appearance of this hole phenomenon, resulting in light transmission. An obvious strip-like phase appears in the matrix, which analysis shows to be a pearlite structure, belonging to a mixed structure of ferrite and cementite. The grain boundaries of TaC particles in the TaC reinforcement layer that are closer to the substrate are less obvious, while the grain boundaries of TaC particles that are far away from the substrate are obvious but have a smaller size and no obvious orientation.

Therefore, based on microstructure analysis and characterization, a tantalum carbide diffusion reaction layer on the surface of GCr15 was successfully designed and prepared at 1100 °C for 1 h. As the thickness of the diffusion-reaction layer increases, the diffusion reaction layer exhibits a gradient growth structure in grain size and morphology. In addition, a large amount of residual stress exists at the interfaces at both ends of the tantalum carbide diffusion reaction layer on GCr15, resulting in a small number of holes during ion thinning, and there are many dislocation entanglements between TaC/TaC particles in the reinforcement layer.

## 4. Conclusions

A tantalum carbide-reinforced steel-based surface reinforcement layer was prepared on the surface of a steel-based alloy using a conventional diffusion reaction heat treatment process through a diffusion-controlled in situ reaction. The following conclusions were obtained:

(1) The phase group of the Ta sample is Ta, TaC, Ta_2_C, and α- Fe, and carbide grains are oriented. The crystal plane orientation of Ta and TaC is {001}. After Ta reacts with C atoms, the crystal orientation of Ta changes from [111] to [101] in the X direction, while the crystal orientation of Ta and TaC does not change to [122] in the Y direction. In the Z direction, the crystal orientation of Ta changes from [112] to [313]. The grain size around the TaC-strengthened layer is mainly concentrated at less than 0.2 μm, wherein Ta grain size is 0~0.2 μm, Ta_2_C grain size is 0~0.2 μm, and TaC grain size is widely distributed between 0 and 0.4 μm. The angular deflection of grain boundaries around the strengthening layer is not obvious. The content of TaC grains at high-angle grain boundaries is equivalent to that at low-angle grain boundaries. Most Ta_2_C grains are at high-angle grain boundaries, while most Ta grains are at low-angle grain boundaries.

The reason is that during the diffusion reaction movement of Ta and C atoms under external forces, the nucleation driving force causes grain growth as well as the strengthening effect of grain boundaries at different angles.

(2) The TaC sample has large residual stress, causing local lattice distortion in the sample. A large amount of residual stress exists at the interfaces at both ends of the tantalum carbide diffusion reaction layer on GCr15, resulting in holes during ion thinning, and there are more dislocation entanglements between TaC/TaC particles in the reinforcement layer. This conclusion can be obtained from the analysis of curling deformation and cavity in the ion thinning process of the FIB sample.

(3) The TEM results show that there are TaC and Ta_2_C phases in the reinforcement layer, and Ta_2_C is dispersed in the diffusion reaction layer, which easily causes the splitting phenomenon of the reinforcement layer, thereby reducing the strength of the reinforcement layer.

## Figures and Tables

**Figure 1 materials-16-03790-f001:**
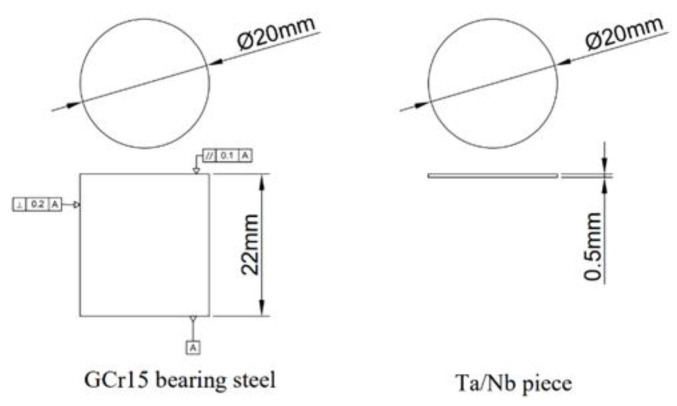
Processing diagram of GCr15 steel and tantalum/niobium plate.

**Figure 2 materials-16-03790-f002:**
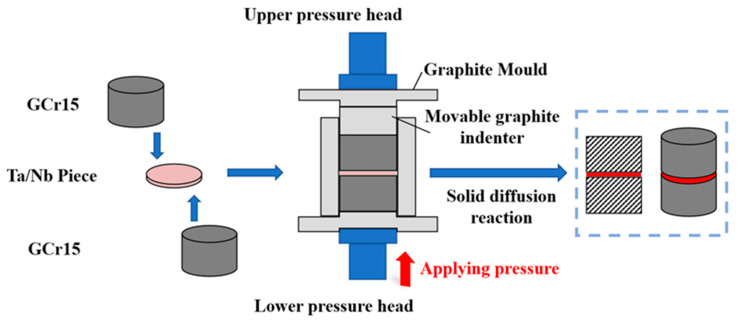
Solid diffusion in situ reaction preparation process schematic diagram.

**Figure 3 materials-16-03790-f003:**
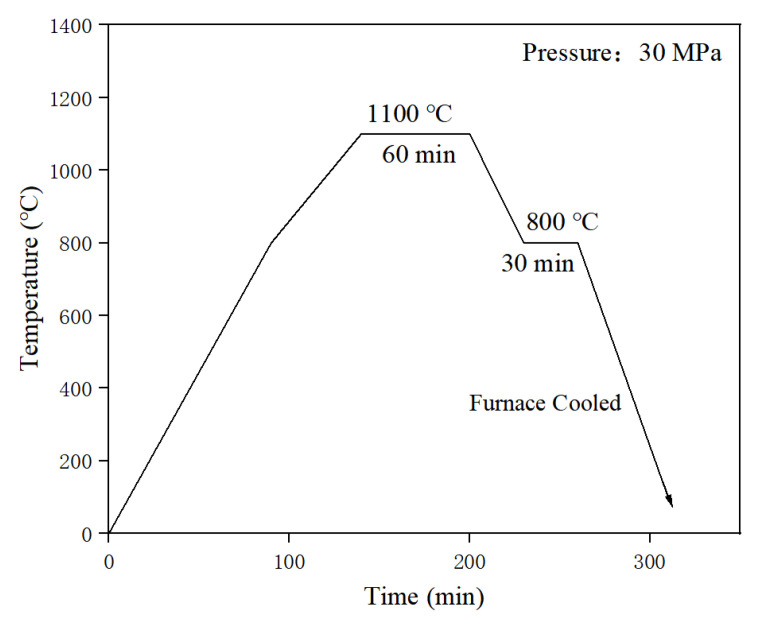
TaC in situ reaction process curve.

**Figure 4 materials-16-03790-f004:**
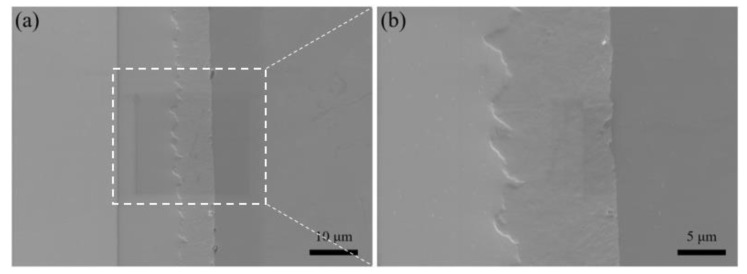
Diagram of EBSD morphology of a selected area of the Ta sample. (**a**) FIB sampling area (**b**) Regional magnification.

**Figure 5 materials-16-03790-f005:**
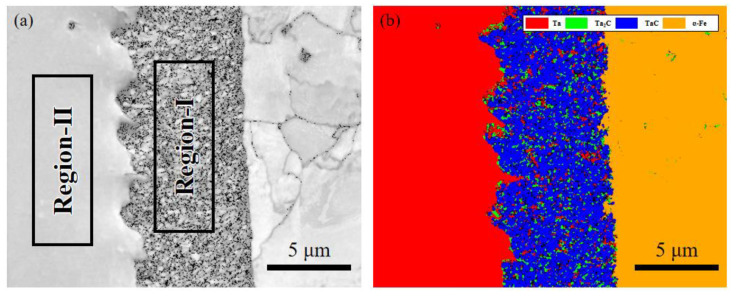
EBSD morphology and phase distribution of Ta samples: (**a**) BSE morphology; (**b**) Phase distribution diagram.

**Figure 6 materials-16-03790-f006:**
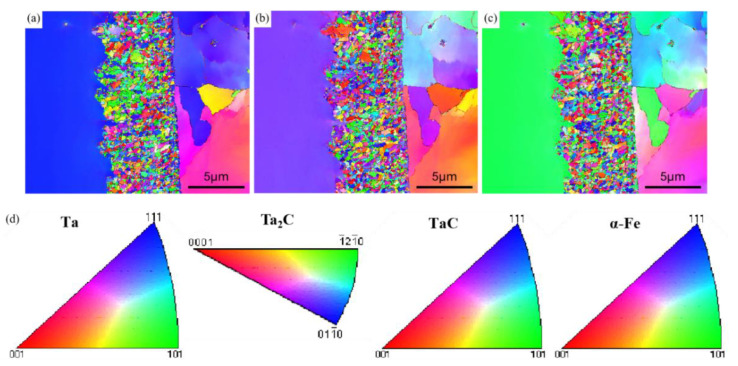
Ta Sample EBSD grain distribution diagram in the (**a**) X direction (**b**) Y direction (**c**) Z direction; (**d**) Antipole diagram.

**Figure 7 materials-16-03790-f007:**
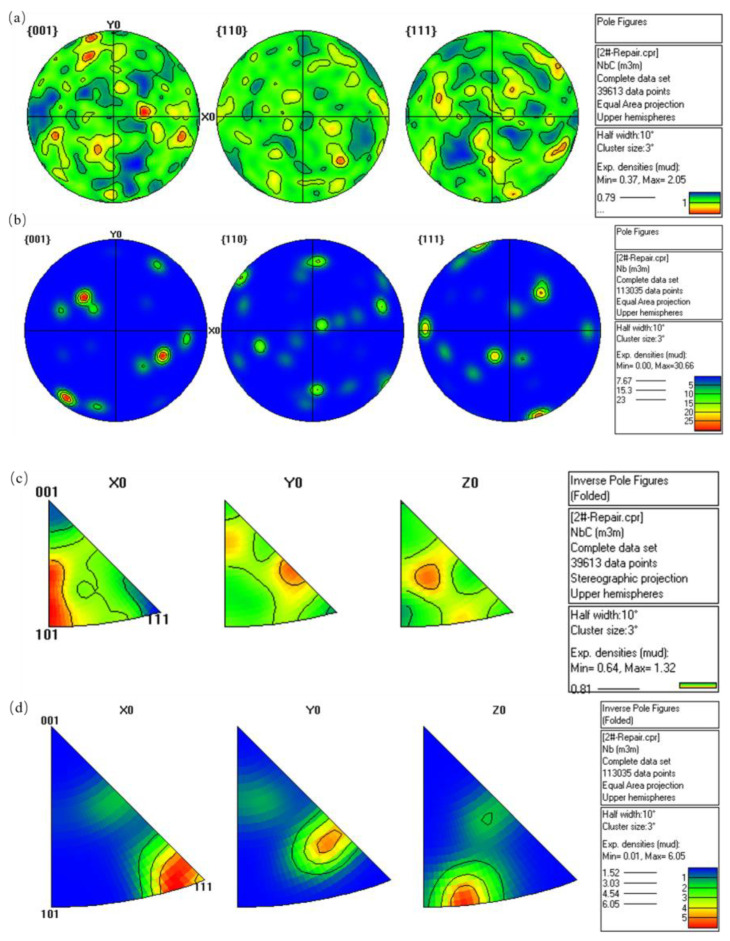
Ta sample: (**a**,**b**) are the pole figures (PF) of Region-I and Region-II; (**c**,**d**) are the inverse pole figure (IPF) maps of Region-I and Region-II.

**Figure 8 materials-16-03790-f008:**
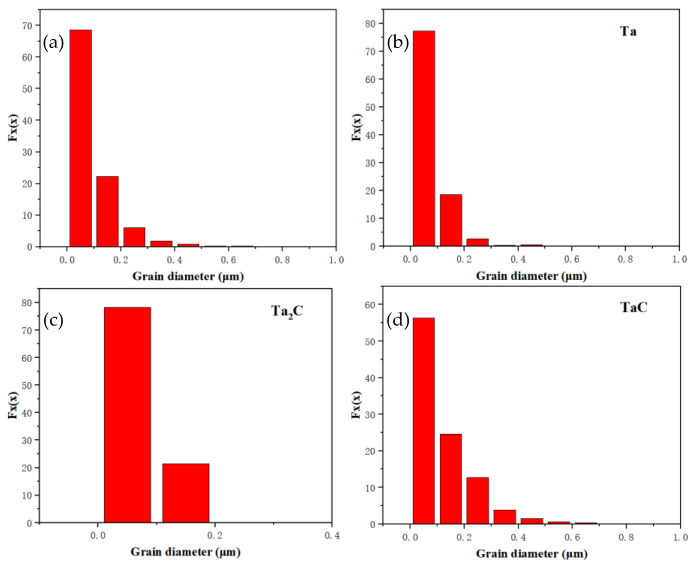
The grain size distribution histogram of the Ta sample: (**a**) all grains, (**b**) Ta, (**c**) Ta_2_C and (**d**) TaC.

**Figure 9 materials-16-03790-f009:**
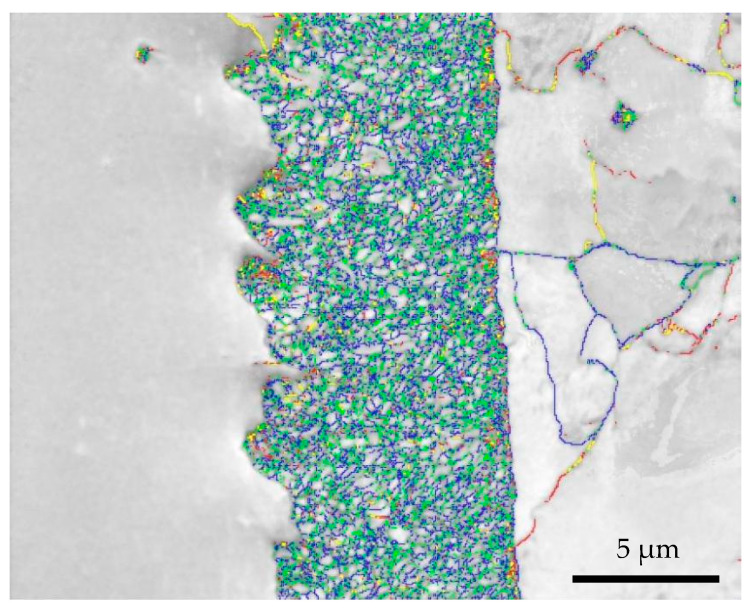
EBSD grain boundary distribution map of Ta sample (red line: 0~2°, yellow line: 2~5°, green line: 5~10°, blue line: 10~15°).

**Figure 10 materials-16-03790-f010:**
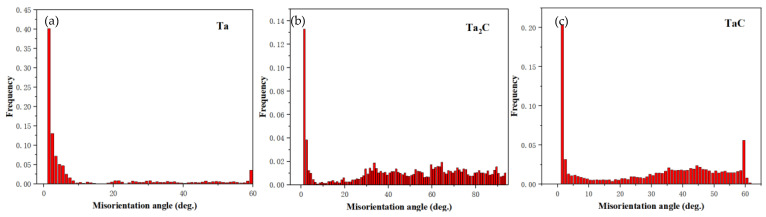
Grain boundary deflection angle distribution of TaC sample: (**a**) TaC, (**b**) Ta_2_C and (**c**) Ta.

**Figure 11 materials-16-03790-f011:**
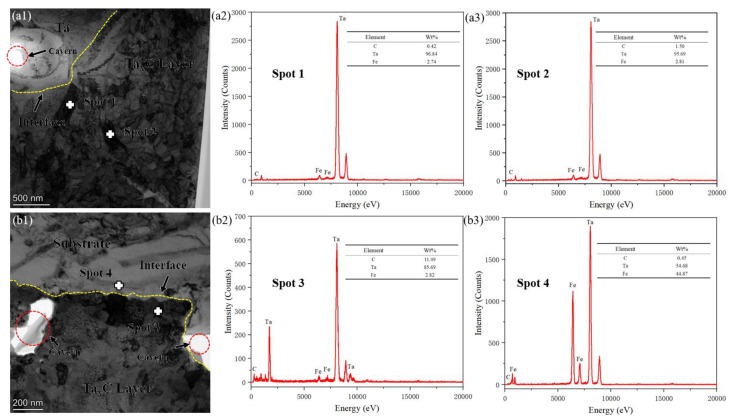
The tantalum carbide reinforced layer on GCr15 prepared at 1100 °C for 1 h: (**a_1_**) TEM image of the interface between the tantalum plate and the reinforced layer; (**a_2_**,**a_3_**) Point scan results corresponding to Spot 1 and Spot 2; (**b_1_**) TEM image of the interface between the reinforced layer and the substrate; (**b_2_**,**b_3_**) Point scan results corresponding to Spot 3 and Spot 4.

**Figure 12 materials-16-03790-f012:**
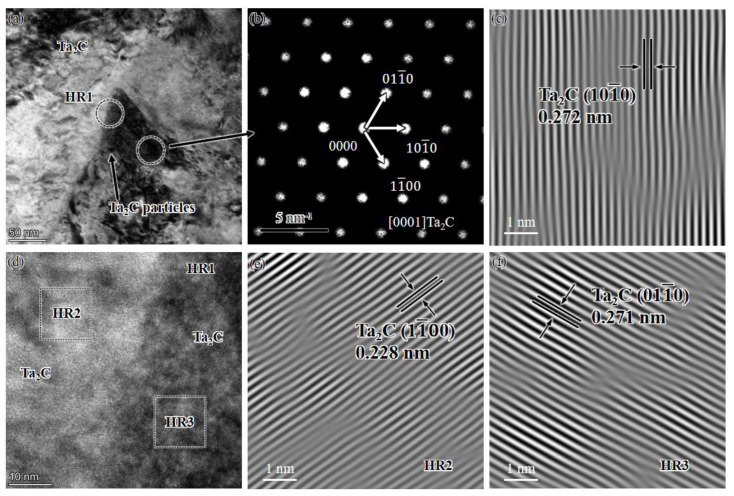
TEM characterization of the interface between tantalum and coating: (**a**) TEM image; (**b**) and (**c**) electron diffraction pattern and high-resolution transmission electron microscope (HRTEM) image obtained with fast Fourier transformation (FFT); (**d**–**f**) HRTEM image of the interface between Ta_2_C particles.

**Figure 13 materials-16-03790-f013:**
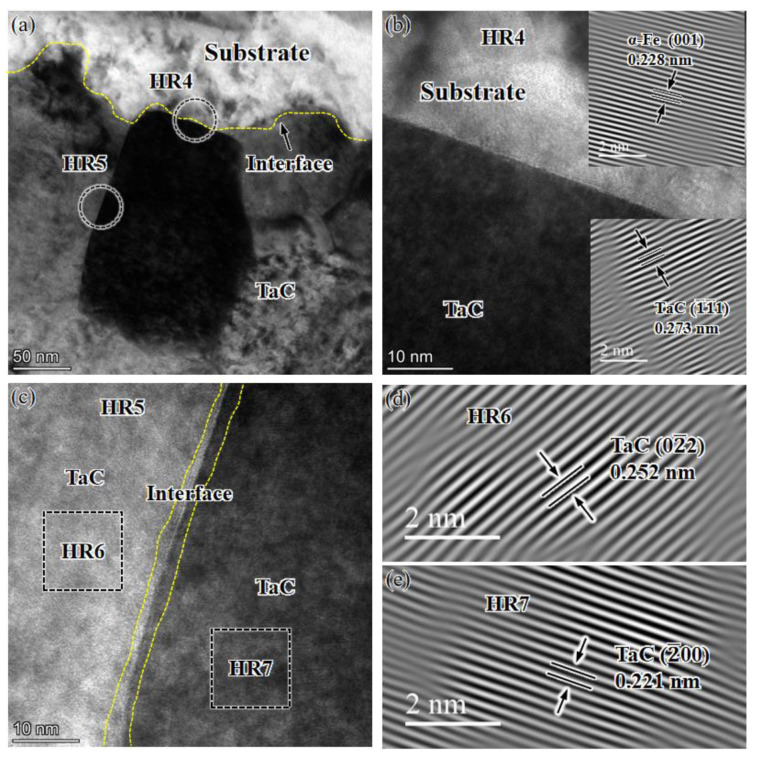
TEM characterization of the interface between the TaC-reinforced layer and the substrate: (**a**) TEM image; (**b**) HRTEM image of the TaC/substrate interface; (**c**–**e**) HRTEM image of the interface between TaC particles.

**Figure 14 materials-16-03790-f014:**
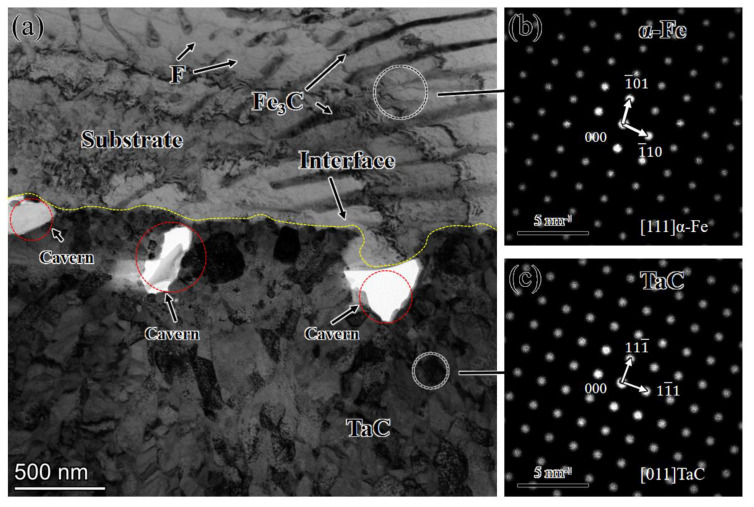
TEM characterization of the interface between the coating and the substrate: (**a**) TEM image; (**b**,**c**) electron diffraction pattern obtained with fast Fourier transformation (FFT).

**Table 1 materials-16-03790-t001:** Chemical composition of GCr15,tantalum plate (mass fraction wt.%).

Materials	C	Si	Mn	Cr	Ni	Mo	Cu	Fe	Ta
GCr15	0.93	0.22	0.61	1.70	0.17	0.22	0.19	Bal.	-
Ta plate	-	-	-	-	-	-	-	-	99.9

**Table 2 materials-16-03790-t002:** Crystallographic parameters for some carbides in the system [24].

	Atomic Radius (Ǻ)	Molecular Weight	Space Group	Maximum Solubility in α-Fe wt. (%)	Maximum Solubility in γ-Fe wt. (%)	Crystal Structure	
						Type	Cell Parameter
Ta	2.09	-	Im-3m (229)	Finite	Finite	bcc	a = b = c = 330.58 pm α = β = γ = 90°
Ta_2_C	-	374	P-3ml (164)	-	-	hcp	a = b = 310.46 pm c = 494.44 pm α = β = 90° γ = 120°
TaC	-	193	Fm-3m (225)	-	0.5–1.0 (1250 °C)	fcc	a = b = c = 445.47 pm α = β = γ = 90°
C	0.91	-	-	0.0218	2.11	-	-
α-Fe	1.430	-	Im-3m (229)	-	-	bcc	a = b = c = 286.60 pm α = β = γ = 90°
γ-Fe	1.425	-	Fm-3m (225)	-	-	fcc	a = b = c = 365.99 pm α = β = γ = 90°

Note: fcc = face-centered cubic close-packing structure; bcc = body-centered cubic close-packing structure; hcp = hexagonal close-packing structure; - = not identified.

## Data Availability

Only the data of this paper is available.

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
