# Peer review of "Study on the Interface Microstructure of TaC/GCr15 Steel Surface Reinforced Layer Formed by In-Situ Reaction"

_materials, 2023, doi:10.3390/ma16103790_

Round 1

Reviewer 1 Report

This work provides an interesting study on the formation of TaC reinforcement layer on a GCr15 steel surface. The characterization study of the layers formed upon the application of heat treatment using a thin Ta sheet is complete and the extracted results and discussion are scientifically sound. Some minor comments:

- The experimental methodology to form the TaC layer is adequate to make the study of the Ta diffusion and C formation in the steel. It would be interesting, however, to mention if the process can be industrially extrapolated or if there would be differences that should be taken into consideration.

- Explain how the heat treatment conditions were selected. It might seem arbitrary, but I assume there is some background behind it.

- Make sure that all the experimental procedures are correctly explained in the corresponding section. Do not repeat or mention experimental procedures in the result section.

- Give more details about the EBSD measurement conditions and specify te scan step size.

- In your conclusions, you mention features that in principle seem negative like residual stresses and Ta2C is dispersed in the reaction-diffusion layer. Try to discuss and elaborate more on the implication of these drawbacks on the development of these coatings in this particular steel. You discard the use of Ta, is it something it can be amended by changing the process conditions?

Author Response

Dear Editors:

We sincerely thank you for your invaluable comments and suggestions on our manuscript entitled “Study on the interface microstructure of TaC / GCr15 steel sur-face reinforced layer formed by in-situ reaction”( materials-2343702). We appreciate the time and efforts that you dedicated to providing feedback on our manuscript and are grateful for the insightful comments on and valuable improvements to our paper. We have carefully made corrections which we hope meet with approval.

detailed information see attachment

Sincerely yours,

Prof. Jilin Li

Apr.17 2023

Reviewer 2 Report

The authors performed hot compression tests and used EBSD and TEM to characterize the microstructure. The presentation is clear and English is easy to understand. However, some major issues are needed for attention.

1.       It is not clear why the authors chose 1100c and 60min.

2.       It is not clear why 800c and 30 min and final cooled samples were not characterized.

3.       The measured 2.72 pm Lattice parameter does not match with the theoretical one, 3.11 pm. The difference is as much as 14%. Why so big difference? Is the particle strength enough to accommodate such large strain, the authors should take this into consideration.

4.       STEM-EDX measurement does not agree with the composition of Ta2C and TaC. Please explain or re-perform the EDX experiments.

5.       Dislocation entanglement is not clear; please take dislocation images under two beam conditions.

6.       Conclusion1: Why there is a rotation of crystal orientation. Please explain.

7.       Conclusion2. Usually, the measurement of residual stress can be done by x-ray, or locally TEM lattice distortion, or HREBSD GND. However, it is not clear to the reviewer how this statement was concluded.

8.       The reviewer suggests performing a tension test along the compression direction to test the coating layer strength and fracture behavior and add a section of mechanical tests to make it as a complete work.

Author Response

(The authors gave the same response as above.)
